# Whole-Body MRI for the Detection of Recurrence in Melanoma Patients at High Risk of Relapse

**DOI:** 10.3390/cancers13030442

**Published:** 2021-01-25

**Authors:** Yanina J. L. Jansen, Inneke Willekens, Teofila Seremet, Gil Awada, Julia Katharina Schwarze, Johan De Mey, Carola Brussaard, Bart Neyns

**Affiliations:** 1Medical Oncology, UZ Brussel, Laarbeeklaan 101, 1090 Brussels, Belgium; gil.awada@uzbrussel.be (G.A.); juliakatharina.schwarze@uzbrussel.be (J.K.S.); bart.neyns@uzbrussel.be (B.N.); 2Medische Beeldvorming, UZ Brussel, 1000 Brussels, Belgium; inneke.willekens@uzbrussel.be (I.W.); johan.demey@uzbrussel.be (J.D.M.); carola.brussaard@uzbrussel.be (C.B.); 3Service de Dermatologie et Vénéréologie, Lausanne University Hospital|CHUV, CH-1011 Laussanne, Switzerland; teofila.caplanusi@chuv.ch

**Keywords:** whole-body MRI, screening, melanoma

## Abstract

**Simple Summary:**

Treatment for metastatic melanoma patients has significantly improved in the last decade. Literature suggests a better prognosis for patients with minimal disease burden. No consensus exists on the surveillance of melanoma patients at high risk of recurrence. In this trial we evaluated the use of whole-body MRI for the surveillance of melanoma patients, more specifically in patients with stage IIIb/c disease following surgical resection and in patients obtaining a durable response on systemic therapy. Overall, we can conclude that whole- body MRI is a safe and sensitive technique for the discovery of distant melanoma metastasis.

**Abstract:**

*Introduction:* No standard protocol for surveillance for melanoma patients is established. Whole-body magnetic resonance imaging (whole-body MRI) is a safe and sensitive technique that avoids exposure to X-rays and contrast agents. This prospective study explores the use of whole-body MRI for the early detection of recurrences. *Material and Methods:* Patients with American Joint Committee on Cancer Staging Manual (seventh edition; AJCC-7) stages IIIb/c or -IV melanoma who were disease-free following resection of macrometastases (cohort A), or obtained a durable complete response (CR) or partial response (PR) following systemic therapy (cohort B), were included. All patients underwent whole-body MRI, including T1, Short Tau Inversion Recovery, and diffusion-weighted imaging, every 4 months the first 3 years of follow-up and every 6 months in the following 2 years. A total body skin examination was performed every 6 months. *Results:* From November 2014 to November 2019, 111 patients were included (four screen failures, cohort A: 68 patients; cohort B: 39 patients). The median follow-up was 32 months. Twenty-six patients were diagnosed with suspected lesions. Of these, 15 patients were diagnosed with a recurrence on MRI. Eleven suspected lesions were considered to be of non-neoplastic origin. In addition, nine patients detected a solitary subcutaneous metastasis during self-examination, and two patients presented in between MRIs with recurrences. The overall sensitivity, specificity, positive predictive value, negative predictive value, and accuracy were, respectively, 58%, 98%, 58%, 98%, and 98%. Sensitivity and specificity for the detection of distant metastases was respectively 88% and 98%. No patient experienced a clinically meaningful (>grade 1) adverse event. *Conclusions:* Whole-body MRI for the surveillance of melanoma patients is a safe and sensitive technique sparing patients′ cumulative exposure to X-rays and contrast media.

## 1. Introduction

The overall 5-year relapse-free survival (RFS) for resected stage IIIA, IIIB, and IIIC patients is 63%, 32%, and 11%, respectively, with the majority of melanoma recurrences and second primary melanomas occurring within 3 years after the initial treatment [1]. Up to 51% of the patients will experience a systemic relapse as a first relapse. The other half will experience local/in-transit metastasis (28%) or regional nodal (21%) metastasis. [1,2] Only 31% of the patients will have an asymptomatic metastasis found on radiologic testing. Current adjuvant therapy with anti-PD-1 antibodies or targeted therapy in BRAF V600 mutant melanoma has improved 3-year RFS in high-risk stage III melanoma to 43–58% [3,4].

Even when patients follow rigorous surveillance protocols, up to 60% will detect their own first relapse [5]. Follow-up guidelines vary from frequent visits for the first 3 years (3–4 monthly) with a lower frequency in the 2 years following (6 monthly to yearly) to only yearly skin examination.

Before 2011, no systemic treatment existed that improved the overall survival (OS) in advanced melanoma patients. The development of immunotherapy and BRAF-targeted therapy led to impressive results in progression-free survival (PFS) and OS. Retrospective data suggest that the burden of disease (number of sites, volume) is a prognostic factor for both PFS and OS [6,7,8,9,10,11], for both treatment modalities. This could indicate a need for an earlier detection of relapse with a low burden of disease [12,13,14,15,16]. Knowledge of long-term outcomes, risk of relapse, and site of relapse after treatment discontinuation is evolving [16,17]. Given that this population is new, no guidelines or consensus exist on their follow-up.

The pattern of metastatic spread for melanoma is unpredictable. Therefore, whole-body imaging techniques are required for a proper follow-up. Whole-body MRI with diffusion-weighted imaging (DWI) is an innovative imaging technique that combines the detailed anatomical information from conventional MRI with functional characterization of tissue from DWI [18]. The performance of MRI for melanoma staging was comparable to computer tomography (CT)- and positron emission tomography (PET)/CT [19,20,21,22,23,24] with an overall sensitivity up to 85%, specificity to 87%, a positive predictive value (PPV) of 90%, a negative predictive value (NPV) of 100%, and an accuracy of 78%. Only for lung metastasis was CT more performant. A sensitivity of around 60% was noted for MRI. These results led to the inclusion of whole-body MRI in the guidelines for the follow-up of high-risk melanoma patients by the German Dermatological Society [25] and the Swiss guidelines [26]. However, until the present, no trials on the use of whole-body MRI for the surveillance of high-risk melanoma patients were conducted.

In this study, we evaluated the safety and efficacy of whole-body diffusion-weighted MRI in 107 patients (68 patients in an adjuvant setting and 39 patients after systemic therapy).

## 2. Material and Methods

### 2.1. Study Design and Patients

From November 2014 until November 2019, all patients with American Joint Committee on Cancer Staging Manual (seventh edition; AJCC-7) stages IIIb/c or -IV melanoma who were disease-free following resection of macrometastases (cohort A) and patients in a durable complete response (CR) or partial response (PR) following systemic therapy (immunotherapy or targeted therapy) (cohort B) were included. All patients underwent whole-body MRI, including T1, short Tau Inversion Recovery, and DW imaging, every 4 months the first 3 years of follow-up and every 6 months in the following 2 years. A blood test, including liver chemistry, lactate dehydrogenase (LDH), and C-reactive protein (CRP), was performed on each visit. A total body skin examination by a dermatologist was performed every 6 months. After 5 years, all patients from cohort A were followed by their dermatologist on a yearly base. The follow-up after 5 years for patients in cohort B was dependent on their disease status and determined at the discretion of the treating physician

Key eligibility criteria verified during the screening procedures were: histologically confirmed malignant melanoma, AJCC Stage IIIb/c or stage IV with no evidence of disease on most recent CT or PET/CT imaging following surgery or for Stage IV after systemic therapy: CR or PR for more than 3 years. Exclusion criteria included: contra-indication for MRI (pacemaker, metallic foreign body in eye, recent operation with prosthetic material (<6 weeks)), claustrophobia, and metallic devices implanted such as hip prostheses altering the imaging quality.

This trial was approved by the Institutional Ethics Committee of the UZ Brussel (ClinicalTrials.gov Identifier: NCT02907827). All patients provided a signed informed consent.

### 2.2. Imaging Protocols

All whole-body MRI examinations were performed on a 3 Tesla scanner (MAGNETOM Skyra, Siemens Healthcare, Erlangen, Germany) with parallel radiofrequency transmission and phased-array surface coils. The MRI protocol included 3D TI weighted VIBE sequence, Short Tau Inversion Recovery (STIR) sequences, and diffusion-weighted imaging (DWI). We created a transverse series with the signal intensity of fat (fat-only), only water (water-only), T1 in-phase, and T1 out-of-phase. The 3D T1 series were reconstructed in sagittal images. As T2-sequence, a coronal STIR sequence was used. Transverse DWI were acquired in eight stations (head/neck, thorax, abdomen, pelvis, upper legs, and lower legs) at b = 50 and b = 800 s/mm^2^. They were interpreted with the apparent diffusion coefficient (ADC) images. Post-processing of the eight stacks of images was required to have an excellent overview. These stacks are composed of one volume. This volume was reconstructed so that it could rotate along its cranio-caudal axis.

### 2.3. Imaging Analyses

Two radiologists analyzed each MRI examination. Any clinical decision was based on the consensus of the two readers. The evaluation of the examination was based on morphological characteristics and DWI appearance. General radiological criteria for metastases were areas with a shape suggestive of a tumor, abnormal signal, hyperintensities on DWI, and corresponding ADC values. A lymph node was suspicious if it was round with a shortest diameter ≥10 mm. Lymph nodes <10 mm, but hyperintense on T1 (suggestive of the presence of melanin) were also suspicious [27]. New subcutaneous lesions were detected on the DWI sequences.

### 2.4. Definition and Study Endpoints

The result of a whole-body MRI was defined as true positive (TP) if metastatic disease was detected by the MRI and was confirmed by biopsy, surgical excision, or by PET/CT in case of multiple metastases. MRI finding was defined as true negative (TN) if the MRI was negative and no disease was detected in the following 4 months (on self-examination, additional consultation, or imaging due to symptoms or incidental finding). A false negative (FN) was defined as a negative MRI but with a relapse in the following 4 months. An MRI finding was defined as false positive (FP) if the possibility of metastatic disease was suspected based on active foci on the MRI, leading to biopsy, surgical management, or other radiological imaging not confirming relapse. In all patients with a suspected relapse on MRI, supplementary imaging was performed before having a therapeutic impact.

Clinical evident disease was defined as a disease causing symptoms such as pain, hemoptysis, dyspnea, etc.

Sensitivity was calculated as TP/(TP + FN) × 100, specificity as TN/(TN + FP) × 100, PPV as TP/(TP + FP) × 100, NPV as TN/(TN + FN) × 100 and accuracy as (TN + TP)/(total population) × 100. RFS was defined as the time from the first MRI to the detection of recurrent melanoma. Cancer-specific OS was defined as the time from the first MRI to the disease-specific death of metastatic melanoma.

### 2.5. Statistical Analyses

The starting point for all survival analyses was the date of the first MRI. Kaplan Meier analyses were used to analyze RFS and OS. Deaths from other causes or unknown outcomes were marked as censored observations for cancer-specific survival.

## 3. Results

### 3.1. Patients Demographic

One-hundred and eleven patients were included (Figure 1: consort diagram); four patients had a screen failure. One hundred and seven patients (68 patients in cohort A and 39 patients in cohort B) were followed with whole-body MRI. The baseline characteristics of the study patients are presented in Table 1. Cohort B consisted of one patient obtaining a CR on chemotherapy, four patients obtaining a CR on targeted therapy and 34 patients obtaining a durable response (CR: 33 patients and PR: 1 patients) on immunotherapy. In total, 585 MRIs were performed, 373 in cohort A and 212 in cohort B. Ten (9%) patients (eight (12%) patients in cohort A and two (5%) patients in cohort B) were lost to follow-up. Ten (9%) patients (five (7%) in cohort A and five (13%) in cohort B) preferred not to continue the follow-up with whole-body MRI. The main reason for discontinuation was logistics.

### 3.2. Relapses

After a median follow-up of 32 months (95% CI, 20–45 months), relapses occurred in 26 (24%) patients, 19 (28%) patients in cohort A and seven (18%) patients in cohort B. Four (4%) patients died due to melanoma-related disease (all cohort A (6%)). No new primary melanomas were diagnosed. Median time to recurrence was 12 months (95% CI; 11–13 months, Figure 2). Median time to recurrence in cohort A and cohort B was, respectively, 11 months (95% CI 4–18 months) and 15 months (95% CI; 7–23 months). Mean RFS was 48 months (95% CI 44–53). For cohort A mean RFS was 44 months (95% CI 39–50) and for cohort B 52 months (95% CI 45–59). Median RFS, and median and mean OS could not be estimated in both cohorts due to the low number of events.

Whole-body MRI detected relapses in 15 (14%) patients, 12 (18%) in cohort A and three (8%) in cohort B. Figure 3 demonstrates an example of a relapse detected by whole-body MRI. All patients had an oligometastatic disease (<3 sites) and a normal LDH at the time of relapse. In one patient, two asymptomatic small brain metastases (4 mm) were discovered (Table 2: site of relapse).

In eleven (10%) patients, relapse was detected in between the scheduled whole-body MRIs. In two (2%) patients relapse was detected due to clinical symptoms. One patient (cohort A) presented with symptoms one week before her second MRI (an interval of 4 months). She evolved into a rapidly progressive disease and died two months later. The retrospective evaluation of the previous MRI did not demonstrate any disease. The second patient (Cohort B) presented with hemoptysis 1 month after his whole-body MRI due to a subcarinal lymph node metastasis (necessitating radiotherapy and initiation of targeted therapy). One patient (cohort A), who was considered as “lost to follow-up” during one year (missing out on all planned MRI evaluations), returned to the hospital after experiencing symptoms of his relapse. Since he did not adhere to the study schedule, he was considered lost to follow-up and his relapse was not considered in RFS or whole-body MRI performance evaluation.

In nine (8%) patients, six (9%) in cohort A and three (8%) in cohort B, the first relapse was identified on self-examination (self-palpation of skin metastasis (five patients and two patients, respectively) or enlarged lymph nodes (one in each cohort)). All of which were resectable and smaller than 1 cm. One patient (cohort B) experienced a skin metastasis at the same date as a new lung metastasis on whole-body MRI.

### 3.3. Whole-Body MRI Performance

Eleven (2%) whole-body MRIs (eight (2%) in cohort A and in three (1%) cohort B) were false positive, leading to four ultrasounds, seven PET/CT scans, and one MRI (liver-specific). The suspected lesions were benign lesions including an adrenal adenoma, a bone hemangioma, two biliary cysts, six subcutaneous nodes (inflammatory tissue and scar tissue), and two skin lesions (seroma and scar tissue).

Overall, the sensitivity for the whole follow-up was 58% (95% CI 37–76), specificity 98% (95% CI 96–99), PPV 58% (95% CI 37–76), NPV 98% (95% CI 96–99), and accuracy 98% (95% CI 97–99). When excluding skin metastases from the equation, sensitivity was 88% (59% CI 62–98), specificity 98% (95% CI 96–99), PPV 58% (95% CI 37–76), NPV 100% (95% CI 98–100), and accuracy 98% (95% CI 97–99) (see Table 3). Sensitivity was higher in cohort A than in cohort B (63% versus 43% for all lesions and 92% versus 75% for distant metastases). The overall performance was comparable for cohort A and cohort B.

### 3.4. Outcome after Recurrence for Cohort A

In six (6%) patients, the recurrence was salvaged with local therapy (surgery in five (5%) patients and radiofrequency ablation in one (1%) patient), after which adjuvant treatment was started in two (2%) patients. In three (3%) patients, anti-PD-1 therapy combined with anti-CTLA-4 was started (two patients experienced PD as best objective response (BOR), and one patient had SD). In six (6%) patients, PD-1 monotherapy was started (BOR CR four patients, PR one patient, and SD in one patient). Targeted therapy was started in four patients (BRAF + MEK inhibitor), leading to a CR in three patients and a PR in one patient.

### 3.5. Outcome after Recurrence for Cohort B

No relapses were seen in the patients obtaining a CR after chemotherapy or targeted therapy. In the immunotherapy arm, thirty-three patients had a CR and one patient had a durable PR at the time of elective discontinuation of immunotherapy. The median follow-up after treatment discontinuation was 44 months (95% CI, 37–42). Relapses occurred in seven (18%) patients; immunotherapy was restarted in four (60%) patients, and all but one obtained a new CR after re-initiating immunotherapy. Targeted therapy was started in three (40%) patients, leading to a durable CR in all three patients.

## 4. Discussion

Given the improved treatment outcome of advanced melanoma patients, surveillance practices following successful treatment for high-risk melanoma patients needs to be reviewed. The improvement of systemic therapy has led to a new cohort of patients benefitting from a highly durable disease remission. To date, no recommendations are available for their follow-up [17,28].

After a follow-up of 32 months (95% CI, 20–45 months), 26 (24%) patients in our study experienced disease recurrence, 19 (28%) patients in cohort A and seven (18%) patients in cohort B. In our cohort, 10% of the study patients were lost to follow-up, and 10% of patients preferred not to be followed with an MRI (the patients requested the use of another imaging modality). These numbers are comparable to other studies evaluating follow-up programs [29,30].

Whole-body MRI is a safe and feasible technique without any risk from cumulative radiation exposure. In our cohort, only two (2%) patients were diagnosed with a symptomatic relapse in between scheduled MRIs. All other systemic relapses were diagnosed with whole-body MRI and all patients had oligo-metastatic disease without an elevated LDH at the diagnosis of recurrence. Relapse was salvaged by locoregional therapy (stereotactic radiotherapy or surgery) in six patients. Survival was not used as an endpoint in this study, so no conclusions or claim can be made on the impact on overall survival. Previous studies on follow-up in melanoma patients could not demonstrate a survival benefit [31]. However, data from trials on both targeted therapy and immunotherapy indicate that patients with a lower volume of disease and a normal LDH have a higher chance of responding to therapy, supporting the potential benefit for the early detection of melanoma recurrence [6,16,28]. An additional major benefit of whole-body MRI is the possibility to identify small, asymptomatic brain lesions. For stage IV melanoma, the presence of brain metastasis has a significant impact on prognosis. With contemporary combination immunotherapy, the outcome may not be different when brain metastases are treated when still asymptomatic [32,33,34].

Nine patients detected a new nodule during self-examination. However, all these lesions were small and resectable at diagnosis. For melanoma patients, a complete skin check and appropriate skin examination training is crucial for detecting locoregional disease. While sensitivity for the detection of all melanoma recurrence is insufficient (58%), the combination of a full skin-examination with a whole-body MRI leads to a sensitivity of 88%, making whole-body MRI an appropriate complementary screening tool for the diagnosis of distant melanoma metastasis. Whole-body MRI had an overall good performance, that was comparable to data in metastatic disease [19,20,22,24,35]. In our study, the PPV (58%) was lower than in previous trials due to 11 patients with a suspected lesion, which were negative on additional imaging. The NPV (98–100%) across all cohorts was high for the detection of relapses, providing a reassurance for patients with a negative scan. A recent study by Turner demonstrated a sensitivity of 79% and a specificity of 88% for the detection of distant metastases in stage III melanoma by CT or PET/CT surveillance. Given the retrospective nature of their trial, the difference in screening intervals, and the wide CI (95%) for sensitivity for all three imaging techniques, no hard conclusions can be drawn about the best imaging technique for screening (PET/CT, CT, or whole-body MRI) [36]. However, with a sensitivity of 88% and a specificity of 98% for the detection of distant metastases in our cohorts and a low rate of false positive compared to PET/CT and CT (2% versus 36%), whole-body MRI should be considered as a new attractive imaging method for future prospective trials on surveillance of melanoma patients, mitigating the risk of secondary cancers induced by ionizing radiation.

To our knowledge, this is the first trial suggesting a surveillance protocol for patients obtaining a durable response on systemic therapy. This is a new patient population and little is known on their risk of relapse and their optimal surveillance. After a median follow-up of 44 months, relapses occurred in seven (21%) patients. The re-initiation of immunotherapy led to a new CR in three out of four treated patients. The initiation of targeted therapy led to a durable CR in all three treated patients. Even though the numbers are small, these responses are encouraging and might indicate the need for early discovery of relapse after immunotherapy discontinuation.

The approximate effective radiation dose of PET/CT is 25 mSV, with a cumulative dose of 325 mSV (in first 3 years, PET/CT every 4 months, followed by two PET/CTs up to 5 years follow-up). Many melanoma patients are relatively young. Especially in patients who obtained a durable response to systemic treatment, a high cumulative radiation dose is usually administered due to staging CT scans and repetitive imaging for treatment response assessments. Additional radiation has to be avoided when a favorable outcome has been achieved, as the cumulative dose imposes a risk of secondary malignancies [37]. One of the disadvantages of whole-body MRI is the acquisition time of 30–60 min (acquisition time in PET/CT is 10–15 min) and the time needed for evaluation of the images by a dedicated radiologist. However, the fluorodeoxyglucoseadministration and uptake will also require the patient to stay in the hospital for over 60 min. In addition, the availability of the MRI scan time is more limited than the availability of the PET/CT.

The strength of this study is that, to the best of our knowledge, it is the first prospective trial evaluating whole-body MRI for the surveillance of melanoma patients at high risk for relapse, with a median follow-up time of three years, only 10 (9%) patients lost to follow-up, and evaluating a total of 107 patients. There are two major weakness of this trial. The first is the lack of a control arm no randomization was foreseen; and, consequently, an eventual impact on survival cannot be claimed. The second is the lack of data collection on the quality of life and physiological wellbeing of the study patients. Therefore, we can only conclude that whole-body MRI is a promising technique for the long-term surveillance of high-risk melanoma patients, mitigating the risk of late adverse events related to the cumulative exposure to imaging modalities with ionizing radiation.

## 5. Conclusions

With a sensitivity of 88% and a specificity of 98% for the detection of distant metastases, whole-body DWI MRI is a safe and sensitive technique for the surveillance of high-risk melanoma patients, sparing the patients cumulative exposure to X-rays and contrast media. Complementary, patients have to be instructed to adhere to performing a self-skin examination in order to achieve an optimal screening procedure for early detection of disease recurrences.

## Figures and Tables

**Figure 1 cancers-13-00442-f001:**
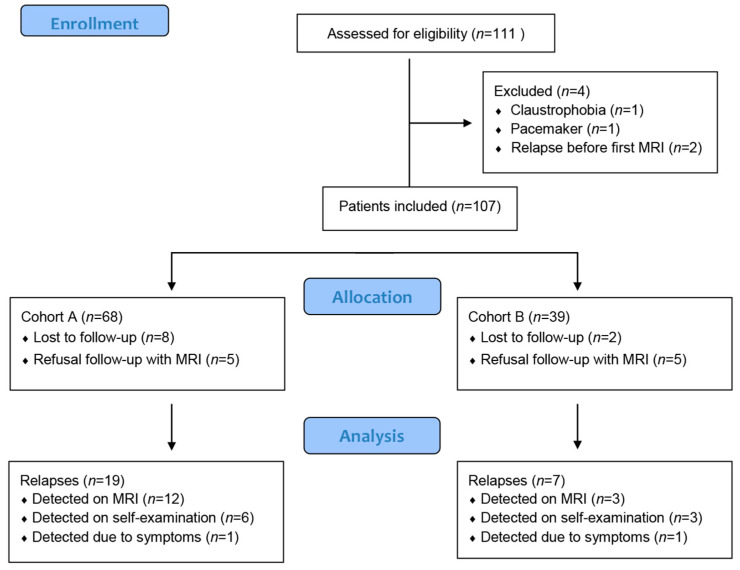
Consort flow Diagram. *n* = number of patients.

**Figure 2 cancers-13-00442-f002:**
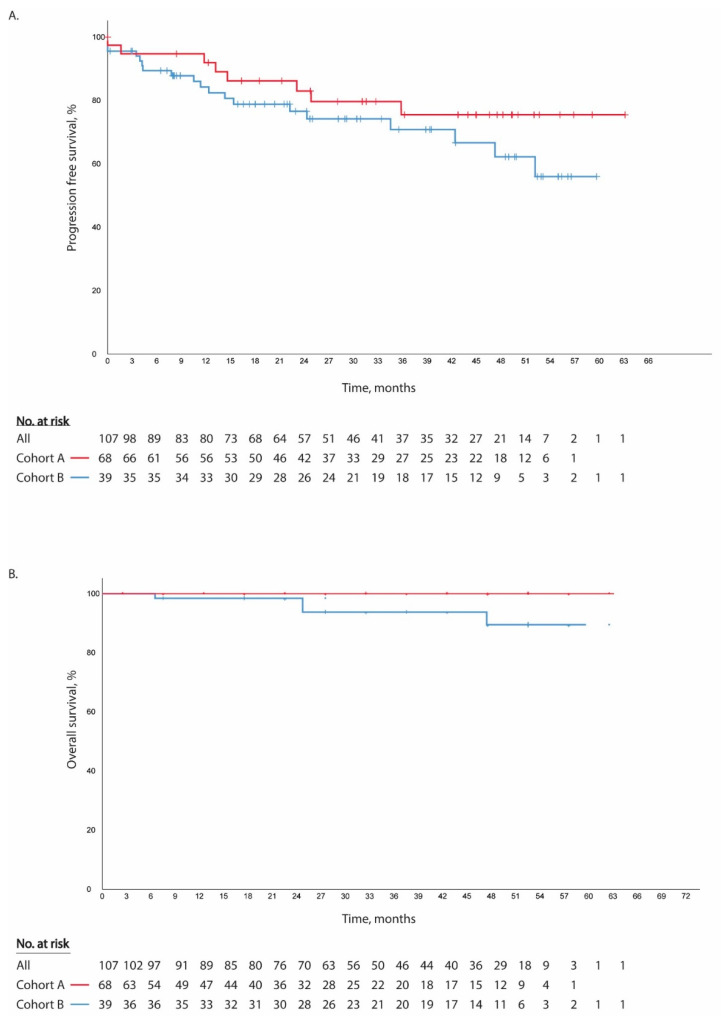
Relapse free survival curves (**A**) and overall survival curves (**B**) for cohorts A and B.

**Figure 3 cancers-13-00442-f003:**
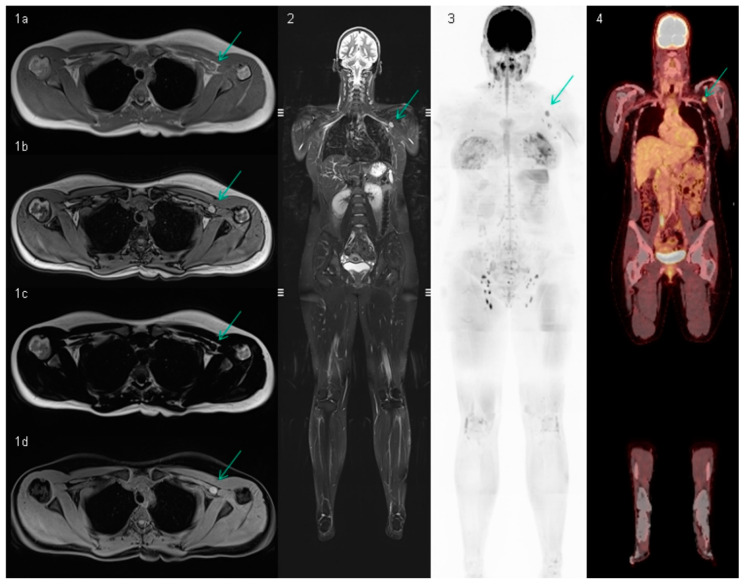
Whole-body MRIof a 32-year-old patient with a first relapse after removing a melanoma on her left shoulder. The images 1a–d are 3D T1 images (VIBE, Siemens, Erlangen), representing an in-phase, out-of-phase, fat-only, and water-only image. The suspected lymph node (10 mm) with a position lateral to the pectoral muscle is indicated with a green flash. Especially in the water-only, the lymph node is somewhat bright (in contrast to other lymph nodes, not visualized on the same image), which might indicate melanin deposition in the lymph node. Image 2, a STIR-image, shows the rounded lymph node as a hyperintense structure. Image 3 is a diffusion-weighted image with a b-value of 800 s/mm^2^. On the inverted grayscale, all lymph nodes are visible as hypo-intense structures. The lymph node to the left of the pectoral muscle is larger and more round than the other lymph nodes. For the decision of a suspect lymph node on whole-body MRI, all sequences must be taken into account. Image 4 is the PET/CT scan of the same patient performed 14 days after the whole-body MRI. The hypermetabolic lymph node on the left is obvious. The metastasis is histologically proven.

**Table 1 cancers-13-00442-t001:** Demographic characterics of participants. Data are expressed as *n*(%) unless otherwise specified. AJCC: American Joint commitee on Cancer 7th edition.

Variable	Cohort A *n* = 68	Cohort B *n* = 39
Total (male/female)	68 (35/33)	39 (17/22)
Median age-years (range)	58 (28–99)	57 (31–85)
Primary site		
Extremities	37 (54%)	8 (21%)
Trunk	12 (18%)	8 (21%)
Head and neck	6 (9%)	3 (8%)
Acral	1 (1%)	
Uveal	2 (3%)	
Unknown primary	10 (15%)	10 (26%)
Ulceration of primary melanoma	20 (29%)	11 (28%)
AJCC Stage		
Ia-IIc	13 (19%)	
IIIa	19 (28%)	
IIIB	12 (18%)	
IIIC	18 (26%)	9 (2%)
IV-M1a	1 (1%)	1 (3%)
IV-M1b		5 (13%)
IV-M1c		18 (46%)
Unknown	5 (7%)	6 (15%)
Prior therapy		
Immunotherapy		
Adjuvant high-dose IFN-α-2b	2 (3%)	8 (21%)
Dendritic-vaccination	4 (6%)	12 (31%)
Mage.A3/AS15 peptide vaccine		
Anti-CTLA-4	9 (13%)	14 (36%)
Anti-PD-1	13 (19%)	5 (13%)
Anti-CTLA-4 + anti-PD-1	3 (4%)	
Targeted therapy		8 (21%)
Dacarbazine		4 (10%)
Temozolomide		1 (3%)
*BRAF^V600^* Mutation		
Yes	39 (58%)	15 (38%)
No	13 (19%)	9 (23%)
Unknown	16 (24%)	15 (38%)

**Table 2 cancers-13-00442-t002:** Site of relapse for cohort A and B. In one patient in cohort A, a lymph node and a lung metastasis was detected synchronous on MRI. Abbreviations: *n* = number of patients, *x* = number of metastatic sites.

	Cohort A (*n* = 68)	Cohort B (*n* = 39)
Median follow-up time (months, 95% CI)	32 (28–36)	34 (28–40)
Recurrence, *n* (%)	19 (28%)	7 (18%)
Detected by MRI	12 (18%)	3 (8%)
Site of recurrence detected on MRI, *x* (%)		
Skin metastases	2 (3%)	
Lymph node	4 (6%)	1 (3%)
Lung	4 (6%)	1 (3%)
Liver	2 (3%)	
Brain	1 (1%)	1 (3%)

**Table 3 cancers-13-00442-t003:** Overview of all identified and classified lesions on MRI, calculation of sensitivity specificity, positive predictive value, negative predictive value and accuracy. Data are expressed as *n*(%) unless otherwise specified. Abbreviations: PPV positive predictive value, NPV negative predictive value, CI confidence interval.

	Cohort A		Cohort B		Total	
	All	Without Skin Metastases	All	Without Skin Metastases	All	Without Skin Metastases
**Relapses**	19	19	7	7	26	26
**True Positive**	12 (3)	12 (3)	3 (1)	3 (1)	15 (3)	15 (3)
**False positive**	8 (2)	8 (2)	3 (1)	3 (1)	11 (2)	11 (2)
**False negative**	7 (2)	1 (1)	4 (1)	1 (1)	11 (2)	2 (1)
**True negative**	346 (93)	352 (94)	201 (95)	205 (97)	548 (94)	557 (95)
**Total # MRI**	373	373	212	212	585	585
**Sensitivity (95% CI)**	63 (39–83)	92 (62–100)	43 (12–80)	75 (22–99)	58 (37–76)	88 (62–98)
**Specificity (95% CI)**	98 (95–99)	98 (95–99)	99 (95–100)	99 (96–100)	98 (96–99)	98 (96–99)
**PPV (95% CI)**	60 (36–80)	60 (36–80)	50 (14–86)	50 (14–86)	58 (37–76)	58 (37–76)
**NPV (95% CI)**	98 (96–100)	100 (98–99)	98 (95–99)	100 (97–100)	98 (96–99)	100 (98–100)
**Accuracy**	96 (94–98)	98 (97–99)	96 (93–99)	98 (96–100)	98 (97–99)	98 (97–99)

## Data Availability

Data available on request due to restrictions, e.g., privacy or ethical reasons. The data presented in this study are available on request from the corresponding author. The data are not publicly available due to the protection of the privacy of research participants.

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
