# Peer review of "Whole-Body MRI for the Detection of Recurrence in Melanoma Patients at High Risk of Relapse"

_cancers, 2021, doi:10.3390/cancers13030442_

Round 1

Reviewer 1 Report

The authors present promising preliminary data supporting the use of MRI as an alternative imaging modality in the surveillance of resected high-risk melanoma, or advanced melanoma after achieving a response to systemic therapy. Importantly, these data do not address the broader question of what clinical benefit any form of surveillance provides in this context, or what the optimal modality and schedule of surveillance should be. In this context it could be made more explicit that these data support inclusion of MRI in future evaluation of surveillance strategies, rather than these data supporting MRI-based surveillance per se. The following corrections/alterations are needed:

Abstract: “MRI triggered no invasive procedures” is a potentially misleading statement, as it appears that MRI findings were followed up with other imaging modalities and all recurrences were confirmed (or excluded) with biopsies, which are by definition invasive procedures.

Consort diagram: the “Relapses” boxes has been duplicated and used for both cohorts. Please correct the data for cohort B.

3.2: There is some interchangeable use of the term “RFS” and “PFS” – please be specific and consistent.

Line 179-182 – it’s not immediately apparent why this patient was considered lost to follow-up. Can this be explained more clearly?

3.3: Table 2 “Relapses” numbers for “Without skin metastases” don’t seem to be correct; for example cohorts A and B had 5 and 2 self-detected skin metastases, respectively.

TP+FN = 18 (A) and 6 (B), each of which is 1 less than the number of stated relapses.

Many of the “total” numbers for TP/FP/FN/TN don’t add up. All data presented in this table need to be inspected for accuracy and several corrections made.

4: Line 223 states 18 (A) and 8 (B) recurrences; previously stated as 19 (A) and 7 (B) in the results. Please correct.

Do the authors have any data regarding recurrences in the patients who declined MRI screening?

The authors correctly point out that MRI was highly sensitive for systemic disease. They also correctly point out that MRI performs poorly for detection of all recurrences (line 243) however this sentence is incomplete (presumably it was supposed to be combined with the sentence beginning line 244) but also this point should be made together with the statement that MRI is highly sensitive for systemic disease. The proportion of recurrences for which MRI was not helpful was substantial (42%) so it is not appropriate to make these points separated by a few paragraphs; these performance issues need to be considered together.

The point about lower burden of disease potentially conferring better outcome from various treatment modalities (lines 236-239) should be concluded with a statement regarding the current state of data supporting a meaningful survival benefit from any kind of screening (clinical, radiological with other modalities), as the point being made is speculative and the uncertainty about a significant impact of screening at all is what underlies the lack of robust surveillance recommendations. This study provides preliminary evidence to support the use of MRI as a surveillance imaging modality, but the authors should be quite clear that the use of any kind of surveillance at all to improve survival outcomes is still not clearly supported by high-level evidence.

Line 254 motivates the preference of MRI over modalities with associated radiation exposure; the authors should however comment briefly on the competing risk of melanoma recurrence which is both very high in this patient population, and far more life-limiting than a putative long-term (e.g. >10-20y latency) risk of secondary malignancy. Apart from individual exceptions (e.g. during pregnancy), the avoidance of radiation exposure in this patient population will not be in its own right sufficient justification for the use of MRI unless MRI is ultimately shown to be cheaper, faster, and/or superior.

Line 256: it would be reasonable to include the FDG administration and uptake time prior to image acquisition time when contrasting modalities – FDG-PET/CT realistically cannot be completed in 10-15 minutes thus if WB-MRI can truly be completed in 30 minutes this is actually favorable to PET/CT.

Several typographical errors need correction including:

192: “scare” tissue

199: “overview off all”

200: in table 2 – positive/negative predictive “factor” should be “value” and used consistently throughout

252: “high” has been repeated

Use “PD-1” consistently, not “PD1”.

Author Response

The authors present promising preliminary data supporting the use of MRI as an alternative imaging modality in the surveillance of resected high-risk melanoma, or advanced melanoma after achieving a response to systemic therapy. Importantly, these data do not address the broader question of what clinical benefit any form of surveillance provides in this context, or what the optimal modality and schedule of surveillance should be. In this context it could be made more explicit that these data support inclusion of MRI in future evaluation of surveillance strategies, rather than these data supporting MRI-based surveillance per se. The following corrections/alterations are needed:

Abstract: “MRI triggered no invasive procedures” is a potentially misleading statement, as it appears that MRI findings were followed up with other imaging modalities and all recurrences were confirmed (or excluded) with biopsies, which are by definition invasive procedures.

We understand this concern and deleted the phrase from the abstract.

Consort diagram: the “Relapses” boxes has been duplicated and used for both cohorts. Please correct the data for cohort B.

Thank you for this comment, this was corrected.

3.2: There is some interchangeable use of the term “RFS” and “PFS” – please be specific and consistent.

This was corrected and RFS was used for all follow up data in our patients.

Line 179-182 – it’s not immediately apparent why this patient was considered lost to follow-up. Can this be explained more clearly?

It concerns this section: One patient (cohort A) returned to the hospital one year after his last whole-body MRI with symptomatic melanoma recurrence (and initiated targeted therapy). This patient was considered lost to follow-up

We altered the section to: One patient (cohort A), who was considered as “lost to follow-up” during one year (missing-out on all planned MRI evaluations), returned to the hospital after experiencing symptoms of his relapse. Since he did not adhere to the study schedule, he was considered lost to follow-up and his relapse was not considered in RFS or whole-body MRI performance evaluation.

3.3: Table 2 “Relapses” numbers for “Without skin metastases” don’t seem to be correct; for example cohorts A and B had 5 and 2 self-detected skin metastases, respectively.

TP+FN = 18 (A) and 6 (B), each of which is 1 less than the number of stated relapses.

Many of the “total” numbers for TP/FP/FN/TN don’t add up. All data presented in this table need to be inspected for accuracy and several corrections made.

The table 2 was designed to give an overview of all detected metastases in all patients (including MRI and additional imaging). We understand the confusion. We adapted the table to only provide an overview of all metastases discovered on MRI. We also modified the legend of the table to clarify the described lesions

Cohort A

(n=68)

Cohort B

(n=39)

Median follow-up time (months, range)

30 (0-60)

26 (0-63)

Recurrence  , n (%)

19 (28%)

7 (18%)

Detected by MRI

12 (18%)

3 (8%)

Site of recurrence detected on MRI, x (%)

    Skin metastases

2 (3%)

    Lymph node

4 (6%)

1 (3%)

    Lung

4 (6%)

1 (3%)

    Liver

2 (3%)

    Brain

1 (1%)

1 (3%)

Table 2: site of relapse for cohort A and B. In one patient in cohort A, a lymph node and a lung metastasis was detected synchronous on MRI. Abbreviations: n = number of patients, x = number of metastatic sites.

We understand that the reviewer is also referring to table 3 (overview of all identified and classified lesions, calculation of sensitivity, specificity, PPV, NPV and accuracy). The numbers of this table were verified and amended when needed.

4: Line 223 states 18 (A) and 8 (B) recurrences; previously stated as 19 (A) and 7 (B) in the results. Please correct.

This was corrected.

Do the authors have any data regarding recurrences in the patients who declined MRI screening?

Besides the patients that were considered lost to follow-up, all patients are still in follow-up at our medical center. Most of them are followed with 4 monthly whole-body MRI or 18F FDG-PET/CT. We agree that this information is interesting, but consider it to be outside of the scope of this manuscript and not to add valuable information with regards to the study data.

The authors correctly point out that MRI was highly sensitive for systemic disease. They also correctly point out that MRI performs poorly for detection of all recurrences (line 243) however this sentence is incomplete (presumably it was supposed to be combined with the sentence beginning line 244) but also this point should be made together with the statement that MRI is highly sensitive for systemic disease. The proportion of recurrences for which MRI was not helpful was substantial (42%) so it is not appropriate to make these points separated by a few paragraphs; these performance issues need to be considered together.

The point about lower burden of disease potentially conferring better outcome from various treatment modalities (lines 236-239) should be concluded with a statement regarding the current state of data supporting a meaningful survival benefit from any kind of screening (clinical, radiological with other modalities), as the point being made is speculative and the uncertainty about a significant impact of screening at all is what underlies the lack of robust surveillance recommendations. This study provides preliminary evidence to support the use of MRI as a surveillance imaging modality, but the authors should be quite clear that the use of any kind of surveillance at all to improve survival outcomes is still not clearly supported by high-level evidence.

We agree with the statement and added the phrase: “Previous studies on follow-up in melanoma patient couldn't demonstrate a survival benefit.”

Nieweg, O. E.; Kroon, B. B. R. The Conundrum of Follow-up: Should It Be Abandoned? Surg. Oncol. Clin. N. Am. 2006, 15 (2), 319–330. https://doi.org/10.1016/j.soc.2005.12.005.

Line 254 motivates the preference of MRI over modalities with associated radiation exposure; the authors should however comment briefly on the competing risk of melanoma recurrence which is both very high in this patient population, and far more life-limiting than a putative long-term (e.g. >10-20y latency) risk of secondary malignancy. Apart from individual exceptions (e.g. during pregnancy), the avoidance of radiation exposure in this patient population will not be in its own right sufficient justification for the use of MRI unless MRI is ultimately shown to be cheaper, faster, and/or superior.

While we understand the point of view of the reviewer, we respectfully disagree. In the patient population we are evaluating 10-year survival rates range between 24-77% (resected stage IV-stage IIIb). In the patient population obtaining a durable response no data is known on their long-term (>10y) survival, but currently available data are encouraging justifying the concern on cumulative radiation dose and subsequent risk for radiation-induced malignancies. The use of new adjuvant treatment options led to significantly improved 3 and 5 year relapse free survival data. This change in prognosis and available treatments for adjuvantly treated melanoma patients requires a re-evaluation of the most appropriate way for the follow-up of this population, including the mitigation of late treatment and imaging related adverse events. Our study was not designed to compare PET CT to MRI or to evaluate the impact on patient survival, however it was designed to evaluate the possible alternative use of whole-body MRI as an effective alternative for imaging modalities that rely on ionizing radiation in this setting. We conclude from our study that Whole-body MRI in combination with a complete skin evaluation, is an effective screening tool that is not associated with the potential long-term hazards known to result from repeated X-ray imaging modalities. We agree that more data on a potential survival impact and comparison to other imaging techniques would be welcome to adapt current follow-up guidelines and imaging modalities. In our opinion, our study results nevertheless positions Whole-body MRI as an attractive alternative with regards of mitigating long-term non-desirable toxicities from ionizing radiation.

Line 256: it would be reasonable to include the FDG administration and uptake time prior to image acquisition time when contrasting modalities – FDG-PET/CT realistically cannot be completed in 10-15 minutes thus if WB-MRI can truly be completed in 30 minutes this is actually favorable to PET/CT.

Thank you for this comment, we added: “However the FDG administration and uptake will also require the patient to stay in the hospital for 60 minutes prior to acquisition of the images.”

Several typographical errors need correction including:

192: “scare” tissue -this is corrected

199: “overview off all” - this is corrected

200: in table 2 – positive/negative predictive “factor” should be “value” and used consistently throughout  - this is corrected

252: “high” has been repeated this is corrected

Use “PD-1” consistently, not “PD1”. This is corrected

Reviewer 2 Report

I do agree with the authors that MRI can detect early brain metastasis rather than conventional CT scan. It would be ideal for all patients with stage 2c or higher patients receive intensive scanning for the follow-up, however, using whole-body MRI for the routine follow-up considered to be not realistic. We cannot spare so many slots only for melanoma follow-up. Moreover, many false-positive cases were observed. I could not agree with the author's conclusion. 

Author Response

I do agree with the authors that MRI can detect early brain metastasis rather than conventional CT scan. It would be ideal for all patients with stage 2c or higher patients receive intensive scanning for the follow-up, however, using whole-body MRI for the routine follow-up considered to be not realistic. We cannot spare so many slots only for melanoma follow-up. Moreover, many false-positive cases were observed. I could not agree with the author's conclusion. 

We agree that the access to MRI may not be trivial in all medical centers and represent a limiting factor with respect to the widespread implementation of our study results. However, the purpose of performing studies is to highlight the scientific value of innovations. These data can serve policy makers to make founded decisions such as investment in a wider availability of MRI for purposes such as reported by our study.

We respectfully disagree that the rate of false-positive cases is high. In our cohort 11 false positive MRI results were observed, out of 585 performed MRIs. A recent study by Turner et al, demonstrated a 36% of false positive findings using (PET) CT imaging. Which is considerable higher than in our study..*

  • * Turner, R. M.; Dieng, M.; Khanna, N.; Nguyen, M.; Zeng, J.; Nijhuis, A. A. G.; Nieweg, O. E.; Einstein, A. J.; Emmett, L.; Lord, S. J.; Menzies, A. M.; Thompson, J. F.; Saw, R. P. M.; Morton, R. L. Performance of Long-Term CT and PET/CT Surveillance for Detection of Distant Recurrence in Patients with Resected Stage IIIA–D Melanoma. Surg. Oncol. 2021. https://doi.org/10.1245/s10434-020-09270-3.

Reviewer 3 Report

 The study is well structured, but can be improved; anyway, the presentation of data is a bit confusing.  Discussion should be also immproved

A point-by-point answer to the following comments should be provided 

  1. Abstract: page 1 line 20: “111 patients were included”; later in the text you declare the patients are 107; 68 patients + 39 patients.

Actually 107 patients were included and 4 were excluded. Please clarify in abstract and later in the text, if deemed necessary.

  1. Page 3 line 108. “MRI finding was defined as true negative (TN) if MRI was negative, and no disease was detected in the following 4 months”. How should the disease be detected in the following 4 months? Please clarify

  1. Please express variables with mean ± SD, when appropriate (i.e. age, follow up,…). Range data can be added to the table beside SD or removed, and should be removed from the text

  1. Figure 1 should be revised “Analysis” part is wrong according to the text. Cohort B data are missing

  1. Page 9 line 203: “In six (6%) patients, the recurrence could be salvaged with local therapy (surgery in 203 five (5%) patients and radiofrequency ablation in one (1%) patient).” I do not understand if these patients WERE treated or COULD be treated. Please clarify

  1. Line 208 “Four patients were started on targeted therapy” please revise this sentence
  2. Line 222 “26 (24%) patients”. The rate should follow the noun, here and throughout the entire text
  3. Line 222 “ After a follow-up of 32 months (range 0-63), 26 (24%) patients experienced recurrent disease, 18 (26%) patients in cohort A and eight (21%) patients in cohort B”

Elsewhere in the text and table you stated that the recurrence were 19 in group A and 7 in group B. please explain such discrepancy and revise it.

  1. Line 223 “Whole-body MRI was highly sensitive for the detection of systemic metastases” report your results in terms of sensitivity also here

  1. Line 226 “These numbers are comparable to other studies evaluating screening.” Provide references

  1. Line 232. “In six patients, metastases were identified that were treatable with locoregional therapy (radiotherapy or surgery)”. syntax should be revised

  1. Line 233 “One of the major benefits of whole-body MRI is the possibility to identify small, asymptomatic brain lesions” The discussion in this point should be expanded. I suggest discussing the following references:

Schwarz D, et al. Susceptibility-weighted imaging in malignant melanoma brain metastasis. J Magn Reson Imaging. 2019 Oct;50(4):1251-1259

Bottoni U, et al. Predictors and survival in patients with melanoma brain metastases. Med Oncol. 2013 Mar;30(1):466. doi: 10.1007/s12032-013-0466-2.

  1. The novel findings should be clearly presented. Moreover, you should compare and discuss your findings with other study/ies performing follow-up with MRI and CT.
  2. Inclusion and exclusion criteria should be added

Author Response

Please see the attachment for details.

The study is well structured, but can be improved; anyway, the presentation of data is a bit confusing.  Discussion should be also improved

We modified the discussion according to your suggestions to:

Given the improved treatment outcome of advanced melanoma patients, surveillance practices following successful treatment for high-risk melanoma patients needs to be reviewed. The improvement of systemic therapy has led to a new cohort of patients benefitting from a highly durable disease remission. To date, no recommendations are available for their follow-up. [17, 28]

After a follow-up of 32 months (range 0-63), 26 (24%) patients in our study experienced disease recurrence, 19 (28%) patients in cohort A and seven (18%) patients in cohort B. In our cohort, 10% of the study patients were lost to follow-up, and 10% of patients preferred not to be followed with an MRI (patient’s request to use another imaging modality). These numbers are comparable to other studies evaluating follow-up programs. [29, 30]

Whole-body MRI is a safe and feasible technique without any risk from cumulative radiation exposure. In our cohort, only two (2%) patients were diagnosed with a symptomatic relapse in between scheduled MRIs. All other systemic relapses were diagnosed with whole-body MRI and all patients had oligo-metastatic disease without an elevated LDH at the diagnosis of recurrence. Relapse could be salvaged by locoregional therapy (stereotactic radiotherapy or surgery) in six patients. Survival was not used as an endpoint in this study, so no conclusions or claim can be made on the impact on overall survival. Previous studies on follow-up in melanoma patient could not demonstrate a survival benefit [29]. However, data from trials on both targeted therapy and immunotherapy indicate that patients with a lower volume of disease and a normal LDH have a higher chance for responding to therapy, supporting the potential benefit for the early detection of melanoma recurrence. [6, 16, 28]. An additional major benefit of whole-body MRI is the possibility to identify small, asymptomatic brain lesions. For stage IV melanoma, the presence of brain metastasis has a significant impact on prognosis. With contemporary combination immunotherapy, outcome may not be different when brain metastases are treated when still asymptomatic. [32, 33, 34]

Nine patients detected a new nodule during self-examination. However, all these lesions were small and resectable at diagnosis. For melanoma patients, a complete skin check and appropriate skin examination training is crucial for detecting locoregional disease. While sensitivity for the detection of all melanoma recurrence is insufficient (58%), the combination of a full skin-examination with a whole-body MRI leads to a sensitivity of 88%, making whole-body MRI an appropriate complementary screening tool for the diagnosis of distant melanoma metastasis. Whole-body MRI had an overall good performance that was comparable to data in metastatic disease. [19,20,22,24,34,35] In our study, the PPV (58%) was lower than in previous trials due to 11 patients with a suspected lesion, which were negative on additional imaging. The NPV (98-100%) across all cohorts was high for the detection of relapse, providing a reassurance for patients with a negative scan. A recent study by Turner, demonstrated a sensitivity of 79% and a specificity of 88% for the detection of distant metastasis in stage III melanoma by CT or PET/CT surveillance. Given the retrospective nature of their trial, the difference in screening intervals and wide 95% CI for sensitivity for all three imaging techniques, no hard conclusions can be drawn about the best imaging technique for screening (PET/CT, Ct or whole-body MRI). [36] However, with a sensitivity of 88% and a specificity of 98% for the detection of distant metastasis in our cohorts and a low rate of false positive compared to PET/CT and CT (2% versus 36%), whole-body MRI should be considered as a new attractive imaging method for future prospective trials on surveillance of melanoma patients, mitigating their risk for secondary cancers induced by ionizing radiation.

To our knowledge, this is the first trial suggesting a surveillance protocol for patients obtaining a durable response on systemic therapy. This is a new patient population and little is known on their risk of relapse and their optimal surveillance. After a median follow-up of 25 months, relapses occurred in seven (21%) patients. The re-initiation of immunotherapy led to a new CR in three out of four treated patients. The initiation of targeted therapy led to a durable CR in all three treated patients. Even though the numbers are small, these responses are encouraging and might indicate the need for early discovery of relapse after immunotherapy discontinuation.

The approximate effective radiation dose of PET-CT is 25mSV, with a cumulative dose of 325mSV (first 3 years PET-CT every 4 months, followed by two PET-CTs up to 5 years follow-up). Many melanoma patients are relatively young. Especially in patients who obtained a durable response to systemic treatment, a high cumulative radiation dose is usually administered due to staging CT-scans and repetitive imaging for treatment response assessments. Additional radiation has to be avoided when a favorable outcome has been achieved as the cumulative dose imposes a risk for secondary malignancies.[37] One of the disadvantages of whole-body MRI is the acquisition time of 30-60 min (acquisition time PET-CT 10-15 min) and the time needed for evaluation of the images by a dedicated radiologist. However, the FDG administration and uptake will also require the patient to stay in the hospital for over 60 minutes. In addition, the availability of the MRI scan time is more limited than the availability of the PET-CT.

The strength of this study is that, to the best of our knowledge, it is the first prospective trial evaluating whole-body MRI for the surveillance of melanoma patients at high risk for relapse; with median follow-up time of three years, only 10 (9%) patients being lost to follow up, evaluating gin total 107 patients. There are two major weakness of this trial. The first is the lack of a control arm (: no randomization was foreseen); and consequently, an eventual impact on survival cannot be claimed. The second is the lack of data collection on the quality of life and physiological wellbeing of the study patients. Therefor we can only conclude that whole-body MRI is a promising technique for the long-term surveillance of high-risk melanoma patient, mitigating the risk for late adverse events related to cumulative exposure to imaging modalities with ionizing radiation.

  1. Conclusions

With a sensitivity of 88% and a specificity of 98% for the detection of distant metastases, whole-body DWI MRI is a safe and sensitive technique for the surveillance of high-risk melanoma patients, sparing the patients cumulative exposure to X-rays and contrast media. Complementary, patients have to be instructed to adhere to performing a self-skin examination in order to achieve an optimal screening procedure for early detection of disease recurrences.

A point-by-point answer to the following comments should be provided 

  1. Abstract: page 1 line 20: “111 patients were included”; later in the text you declare the patients are 107; 68 patients + 39 patients.

Actually 107 patients were included and 4 were excluded. Please clarify in abstract and later in the text, if deemed necessary.

 We added the screen failures in the abstract: 111 patients were included (4 screen failures, cohort A: 68 patients; cohort B: 39 patients)

  1. Page 3 line 108. “MRI finding was defined as true negative (TN) if MRI was negative, and no disease was detected in the following 4 months”. How should the disease be detected in the following 4 months? Please clarify

 We added: “MRI finding was defined as true negative (TN) if MRI was negative, and no disease was detected in the following 4 months (on self-examination, additional consultation or imaging due to symptoms or incidental finding).”

  1. Please express variables with mean ± SD, when appropriate (i.e. age, follow up,…). Range data can be added to the table beside SD or removed, and should be removed from the text

 Standard deviations are added were applicable.

  1. Figure 1 should be revised “Analysis” part is wrong according to the text. Cohort B data are missing

 The consort diagram is corrected

  1. Page 9 line 203: “In six (6%) patients, the recurrence could be salvaged with local therapy (surgery in 203 five (5%) patients and radiofrequency ablation in one (1%) patient).” I do not understand if these patients WERE treated or COULD be treated. Please clarify

We changed the phrase to: In six (6%) patients, the recurrence was salvaged with local therapy (surgery in five (5%) patients and radiofrequency ablation in one (1%) patient), after which adjuvant treatment was started in two (2%) patients).

  1. Line 208 “Four patients were started on targeted therapy” please revise this sentence

We altered the sentence to Targeted therapy was started in four patients (BRAF + MEK inhibitor), leading to a CR in three patients and a PR in one patient.

  1. Line 222 “26 (24%) patients”. The rate should follow the noun, here and throughout the entire text

We will adapt this if required by the editorial board and conform the desired layout of the publisher

  1. Line 222 “ After a follow-up of 32 months (range 0-63), 26 (24%) patients experienced recurrent disease, 18 (26%) patients in cohort A and eight (21%) patients in cohort B”

Elsewhere in the text and table you stated that the recurrence were 19 in group A and 7 in group B. please explain such discrepancy and revise it.

This was corrected and was due to a modification in an earlier version of the manuscript. The data are corrected

  1. Line 223 “Whole-body MRI was highly sensitive for the detection of systemic metastases” report your results in terms of sensitivity also here

 This part of the conclusion was altered to

Nine patients detected a new nodule on self-examination. However, all these lesions were small and resectable at diagnosis. For melanoma patients, a complete skin check and appropriate skin examination training appears to be crucial for early detection of locoregional disease relapses. While sensitivity for the detection of all melanoma recurrence is insufficient (58%), the combination of a full skin-examination with a whole-body MRI leads to a sensitivity of 88%, making whole-body MRI an appropriate screening tool for distant melanoma metastasis. Whole-body MRI had an overall good performance that was comparable to data in metastatic disease.[21,22,24,26,31;32]In our study, the PPV (58%) was lower than in previous trials due to 11 patients with a suspected lesion, which were negative on additional imaging. The NPV (98-100%) across all cohorts was high for the detection of relapse, providing a reassurance for patients with a negative scan. A recent study by Turner, demonstrated a sensitivity of 79% and a specificity of 88% for the detection of distant metastasis in stage III melanoma by CT or PET/CT surveillance. Given the retrospective nature of their trial, the difference in screening intervals and the higher number of relapses, no direct comparison or definitive conclusions can be made about the comparison between PET/CT, Ct or MRI for the surveillance of melanoma patients. However, with a sensitivity of 88% and a specificity of 98% for the detection of distant metastasis in our cohorts, whole-body MRI is a promising technique for the surveillance of melanoma patients.

  1. Line 226 “These numbers are comparable to other studies evaluating screening.” Provide references

 These were added

  1. Line 232. “In six patients, metastases were identified that were treatable with locoregional therapy (radiotherapy or surgery)”. syntax should be revised

We corrected the phrase to:  Relapse could be salvaged by locoregional therapy (stereotactic radiotherapy or surgery) in six patients.

  1. Line 233 “One of the major benefits of whole-body MRI is the possibility to identify small, asymptomatic brain lesions” The discussion in this point should be expanded. I suggest discussing the following references:

Schwarz D, et al. Susceptibility-weighted imaging in malignant melanoma brain metastasis. J Magn Reson Imaging. 2019 Oct;50(4):1251-1259

Bottoni U, et al. Predictors and survival in patients with melanoma brain metastases. Med Oncol. 2013 Mar;30(1):466. doi: 10.1007/s12032-013-0466-2.

An additional major benefit of whole-body MRI is the possibility to identify small, asymptomatic brain lesions. For stage IV melanoma, the presence of brain metastasis has a significant impact on prognosis. With contemporary combination immunotherapy, outcome may not be different when brain metastases are treated when still asymptomatic

  1. The novel findings should be clearly presented. Moreover, you should compare and discuss your findings with other study/ies performing follow-up with MRI and CT.

We refer to our reply on item 9.More specifically to the following paragraph

A recent study by Turner, demonstrated a sensitivity of 79% and a specificity of 88% for the detection of distant metastasis in stage III melanoma by CT or PET/CT surveillance. Given the retrospective nature of their trial, the difference in screening intervals and wide 95% CI for sensitivity for all three imaging techniques, no hard conclusions can be drawn about the best imaging technique for screening (PET/CT, Ct or whole-body MRI). [36] However, with a sensitivity of 88% and a specificity of 98% for the detection of distant metastasis in our cohorts and a low rate of false positive compared to PET/CT and CT (2% versus 36%) , whole-body MRI should be considered for future prospective randomized trials on surveillance of melanoma patients.

  1. Inclusion and exclusion criteria should be added

We thank you for this remark and added the following to study design and patients

Key eligibility criteria verified during the screening procedures were: Histologically confirmed malignant melanoma, AJCC Stage IIIb/c or stage IV with no evidence of disease on most recent CT or PET-CT imaging following surgery or for Stage IV after systemic therapy: CR or PR for more than 3 years . Exclusion criteria included: contra-indication for MRI (pacemaker, metallic foreign body in eye, recent operation with prosthetic material (< 6weken), claustrophobia and metallic devices implanted such as hip prostheses altering the imaging quality

Round 2

Reviewer 2 Report

i have no further comment

Author Response

There are no comments to respond to!

Reviewer 3 Report

Most of the concerns were addressed. Please address the followings

  1. Again, from my first report: figure 1 is wrong and has not been corrected yet. Relapses data have to be corrected because are not consistent with paragraph “3.2 Relapses”
  2. Again, from my first report, the data should be presented as mean ±SD (instead of range), when appropriate, throughout the entire manuscript
  3. Line 263 “Relapse could be salvaged by locoregional therapy (stereotactic radiotherapy or surgery) in six patients.” Do you mean “was salvaged” ? please clarify

Author Response

  1. Again, from my first report: figure 1 is wrong and has not been corrected yet. Relapses data have to be corrected because are not consistent with paragraph “3.2 Relapses”

Figure one was altered and resubmitted, but this altered figures and tables are not shown in the revision. We confirm

In cohort A: 19 relapses, 12 discovered on MRI, one discovered due to symptoms and 6 due to self palpation (12+1+6 =19)

In cohort B: 7 relapses, 3 detected on MRI, One discovered due to symptoms and 3 due to selfpalpation (3+1+3 = 7)

In total : 12+3=15 detected on MRI, 1+1=2 due to symptoms and 6+3=9 on self palpation

  1. Again, from my first report, the data should be presented as mean ±SD (instead of range), when appropriate, throughout the entire manuscript

For age, median (range) is a common accepted way of presentation in our patient cohort 1–3

The relapse section was modified to

After a median follow up of 32 months (95% CI, 20-45 months), relapses occurred in 26 (24%) patients, nineteen (28%) patients in cohort A and seven (18%) patients in cohort B. Four (4%) patients died due to melanoma-related disease (all cohort A (6%)). No new primary melanomas were diagnosed. Median time to recurrence was 12 months (95% CI; 11-13 months, Figure 2). Median time to recurrence in cohort A and cohort B was respectively 11 months (95% CI 4-18 months) and 15 months (95% CI; 7-23 months). Mean RFS was 48 months (95% C.I., 44-53 months), for cohort A mean RFS was 44 months (95% C.I., 39-50 months) and for cohort B mean RFS was 52 months (95% C.I,. 45-59months). Median RFS, median and mean OS could not be estimated in both cohorts due to the low number of events.

(1)           Weber, J.; Mandala, M.; Del Vecchio, M.; Gogas, H. J.; Arance, A. M.; Cowey, C. L.; Dalle, S.; Schenker, M.; Chiarion-Sileni, V.; Marquez-Rodas, I.; Grob, J. J.; Butler, M. O.; Middleton, M. R.; Maio, M.; Atkinson, V.; Queirolo, P.; Gonzalez, R.; Kudchadkar, R. R.; Smylie, M.; Meyer, N.; Mortier, L.; Atkins, M. B.; Long, G. V.; Bhatia, S.; Lebbé, C.; Rutkowski, P.; Yokota, K.; Yamazaki, N.; Kim, T. M.; De Pril, V.; Sabater, J.; Qureshi, A.; Larkin, J.; Ascierto, P. A. Adjuvant Nivolumab versus Ipilimumab in Resected Stage III or IV Melanoma. N. Engl. J. Med. 2017, 377 (19), 1824–1835. https://doi.org/10.1056/NEJMoa1709030.

(2)           Weber, J. S.; D’Angelo, S. P.; Minor, D.; Hodi, F. S.; Gutzmer, R.; Neyns, B.; Hoeller, C.; Khushalani, N. I.; Miller, W. H.; Lao, C. D.; Linette, G. P.; Thomas, L.; Lorigan, P.; Grossmann, K. F.; Hassel, J. C.; Maio, M.; Sznol, M.; Ascierto, P. A.; Mohr, P.; Chmielowski, B.; Bryce, A.; Svane, I. M.; Grob, J. J.; Krackhardt, A. M.; Horak, C.; Lambert, A.; Yang, A. S.; Larkin, J. Nivolumab versus Chemotherapy in Patients with Advanced Melanoma Who Progressed after Anti-CTLA-4 Treatment (CheckMate 037): A Randomised, Controlled, Open-Label, Phase 3 Trial. Lancet Oncol. 2015, 16 (4), 375–384. https://doi.org/10.1016/S1470-2045(15)70076-8.

(3)           Blank, C. U.; Larkin, J.; Arance, A. M.; Hauschild, A.; Queirolo, P.; Del Vecchio, M.; Ascierto, P. A.; Krajsova, I.; Schachter, J.; Neyns, B.; Garbe, C.; Chiarion Sileni, V.; Mandalà, M.; Gogas, H.; Espinosa, E.; Hospers, G. A. P.; Miller, W. H.; Robson, S.; Makrutzki, M.; Antic, V.; Brown, M. P. Open-Label, Multicentre Safety Study of Vemurafenib in 3219 Patients with BRAF V600 Mutation-Positive Metastatic Melanoma: 2-Year Follow-up Data and Long-Term Responders’ Analysis. Eur. J. Cancer 2017, 79, 176–184. https://doi.org/10.1016/j.ejca.2017.04.007.

  1. Line 263 “Relapse could be salvaged by locoregional therapy (stereotactic radiotherapy or surgery) in six patients.” Do you mean “was salvaged” ? please clarify

To be more clear, we modified the phrase to: relapse was salvaged by locoregional therapy (stereotactic radiotherapy or surgery) in six patients
